# Investigating explainable human-robot interaction with augmented reality

Chao Wang
*Honda Research Institute EU*
*Offenbach, Germany*
chao.wang@honda-ri.de

Anna Belardinelli
*Honda Research Institute EU*
*Offenbach, Germany*
anna.belardinelli@honda-ri.de

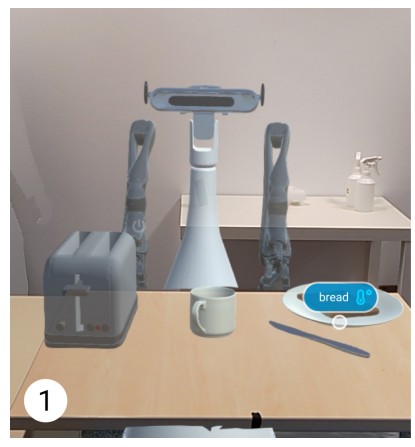 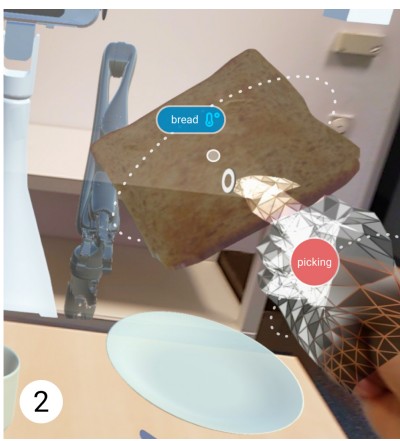 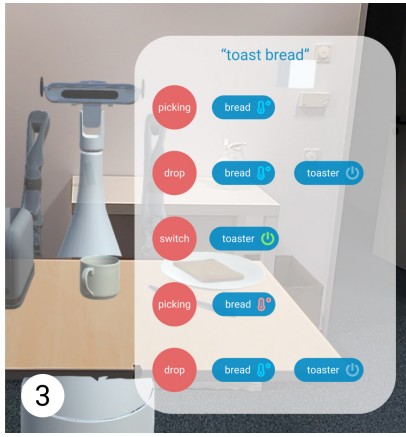

Fig. 1: Teaching by demonstration with XAI feedback. Three layers of information are displayed: 1) Object-related information, e.g., a label, which includes object name (bread) and its state ("cold" icon), appears whenever the human teacher's gaze is on the bread; 2) action-related information, e.g., human teachers "picking" action is shown on the hand of the teacher; and 3) Robot-reasoning information, e.g., the summary of the observed sequence appears after the demonstration.

*Abstract*—In learning by demonstration with social robots, fluid and coordinated interaction between human teacher and robotic learner is particularly critical and yet often difficult to assess. This is even more so if robots are to learn from non-expert users. In such cases, it is sometimes troublesome for the teacher to get a grasp of what the robot knows or to assess if a correct representation of the task has been formed even before the robot demonstrates it back. Here, we introduce a new feedback modality making use of Augmented Reality to visualize the perceptual beliefs of the robot in an interactive way. Such cues are indeed overlaid directly on the shared workspace, as perceived by the teacher, without the need for an explicit inquiry. This allows the teacher to access the robot's situation understanding and adapt their demonstration online, while finally reviewing the observed sequence. We further propose an experimental framework to assess the benefits of such feedback modality - as compared to more established modalities such as gaze and speech - and to collect dyadic data in a quick, integrated, and relatively realistic way. The planned user study will help to assess human-robot coordination across communicative cues and the combination of different modalities for explainable robotics.

*Index Terms*—Explainable robotics, Augmented Reality, human-robot interaction, behavioral user studies

## I. INTRODUCTION

The current spread of social and assistive robotics applications is increasingly highlighting the need for robots that can be easily taught and interacted with, even by users with no technical background. Indeed, even when an autonomous discovery or learning from knowledge bases (e.g., [1], [2]) will be commonplace, specific and personalized tasks will need to be learned by robots in interaction with their primary users. Although some proposed approaches rely on language (e.g., [3]), this modality still represents a major challenge for smooth human-robot interaction (HRI), and learning by demonstration remains the preferred form of interactive learning. Such a teacher-learner scenario has been long explored in HRI, mostly focusing on acquiring human demonstrations via kinesthetic teaching, teleoperation, or passive observation [4], [5]. Still, such approaches will not be suitable for social robots deployed in homes or other human-centered environments. In these situations, learning from a human will be a form of social interaction, with the two partners relying on a model of each other's intents and capabilities. Tutoring is a form of communication where both the tutor and the learner exchange information with multi-modal signals on a social and on a task level [6], via instructions, feedback, and physical actions. Users should receive feedback from robots, so to structure the demonstration according to their representation capabilities, while also being able to review and correct what the robot is learning. This means that there are feedback loops by which both partners shape their behaviors respectively [7].

Such interdependence needs to be investigated in an integrated system to assess the best ways to develop teachable robots [6], [8].

As recently put forward by [9], often the very first obstacle in HRI is that we cannot assume similar perception and situation understanding capabilities as ours when interacting with a robot. Especially during a demonstration, a user needs to be aware of what perceptual beliefs the robot holds at the moment. As a solution, [9] propose that people either observe the robot interacting with the environment to construct a detailed mental model of its functioning or the robot actively signals its perceptual beliefs to guide the tutor. This latter would allow a user to interact with the robot without specific training and observation. In this paper, we propose such a solution based on wearable Augmented Reality (AR) and tailored to teaching scenarios. To assess its effectiveness and compatibility with other established feedback modalities we devised an experimental framework comparing different modalities in isolation and combination. Results from such a user study will shed light on the perspective use of AR and eXplainable Artificial Intelligence (XAI) cues to teach robots new tasks intuitively and effectively, while enhancing trust and acceptance towards robotic systems.

## II. RELATED WORK

With computational improvements and availability of related cheaper hardware in the last decade, the use of AR as a new interface to ease communication between humans and robots has rapidly increased [10]–[12]. Further, the spread of robotics in public scenarios has intensified the need for transparency and legibility, already addressed in HRI (e.g., [13]), leading to the emergence of deeper interpretability instances somehow similar to those of XAI. Within social robotics yet the focus is more on making embodied agents understandable not just in their behavior and decision-making, but more in general in their knowledge, intentions, and perceptions [14]. According to a recent review [15], not many approaches have assessed experimentally the benefits of designs for explainable agency, or testbeds were rather simplistic. Further, while most application scenarios are in the domain of human-robot collaboration, none thus far seems to address a teaching by demonstration scenario where the robot is the learner. As to the perceptual belief problem introduced above, even when the user can assume from previous experience that certain objects in the shared workspace are known to the robot, the robot could still not recognize a specific instance with sufficient confidence because of occlusions or light conditions. The user needs to be timely informed about this, if possible without interfering with their current perception or execution. As suggested by [9], [14], this could be done by gaze cueing or by leveraging new XR possibilities, as in our proposal explained next.

## III. XAI CUES FOR LEARNING BY DEMONSTRATION

During teaching, the human teacher needs to ensure that correct data is fed to the robot. Many interactive machine learning (iML) interfaces assist the human teacher in achieving this goal by clearly showing the collected data [16]–[18]. We target here semantic learning, where a robot needs to collect two types of information: 1) the sequence of actions to complete a certain task, and 2) the object state change due to a certain action. Often AR can enhance HRI by showing robots' internal states [19]–[21]. For the above reasons, 3 layers of information including both collected data and social cues are integrated into the XAI interface design:

- **Object-related information**: A label shows the recognition of an object and its state (Figure 1-1). When the teacher looks at the object, the label pops up.
- **Action-related information**: A label shows the recognized actions of the teacher, such as "pick up", "drop" or "switch on/off" (Figure 1-2), whenever they manipulate an object in the scene.
- **Robot reasoning information**: This layer can show information to the teacher during direct social interaction with the learner, e.g., the action sequence recorded by the robot (Figure 1-3). In the future, more complex reasoning and learning output might be visualized in this way.

## IV. EXPERIMENTAL FRAMEWORK

### A. Proposed experimental design and procedure

To validate the benefits provided by the proposed XAI cues, this modality needs to be compared to more explored, human-like modalities. Gaze cueing and joint attention have been tackled extensively in HRI (see [22] for a review). They belong to the most basic social cues usually implemented in robots, especially in teaching scenarios (e.g., [23], [24]) and have been demonstrated to affect human partners [25]. To make the gaze and XAI channels comparable independently of the information content, the gaze modality is integrated with some utterances. Our assumption here is that gaze is a more human-like feedback modality, hence more familiar to express attention and engagement in an embodied way, still XAI cues are more expressive and offer a more direct insight into the robot's mental state, without the need for the user to actively monitor the robot. Still, these two modalities are by design different in their communicative use and they also target different perceptual channels in the users, hence it might be the case that users find the combination of the two better. Our hypotheses for the experimental study are thus the following:

- **H1**: the XAI modality improves the user experience, coordination and user's awareness of the robot state.
- **H2**: combining XAI with other feedback modalities further improves the experience.

Different modalities are planned to be tested in an AR environment. The user wearing a Head Mounted Display (HMD) will demonstrate simple kitchen tasks to the robot observing the demonstration and reacting to certain cues from the teacher. Fundamentally, three kinds of feedback are conveyed by the robot to the user during the demonstration:

- *social attunement*: the robot sees that the user wants to interact with it and signals it is attending;

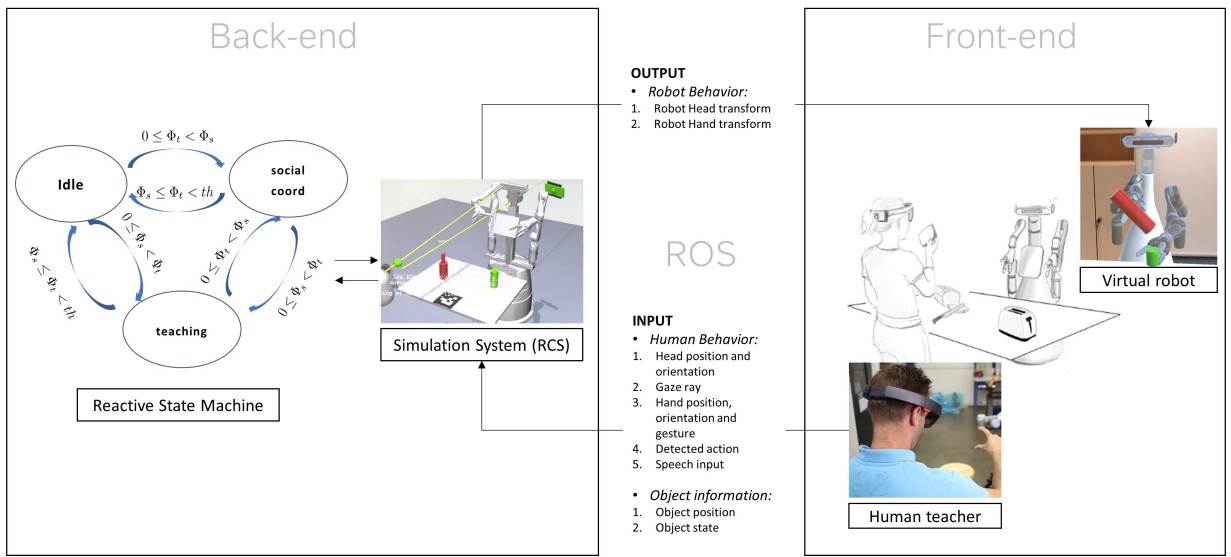

Fig. 2: System architecture. The right block shows the front-end interface with the AR setup, the left block the exchange with the robot simulation and the behavior state machine via ROS.

- *demonstration observation*: the user demonstrates the task and the robot follows by focusing on relevant objects;
- *review*: the robot reports the observed sequence at the end of the demonstration.

To verify our hypotheses, we compare three conditions (factor *modality*): *gaze-speech*, where the robot follows the demonstration orienting its gaze and uttering what it sees; *XAI*, where XAI cues are displayed directly on the attended object holograms; a *combined* condition, expressing feedback through both gaze, speech, and XAI. These behaviors are further described in section IV-B.

In each condition, three different physical tasks will be demonstrated. Envisioning a scenario where a naïve user will teach the robot to prepare or warm up food, we considered for variability in the demonstrations the following tasks (factor *task*): toasting bread in the toaster (*easy*), heating up milk in the microwave (*medium*), boiling eggs in a pot on the stove (*difficult*). Please note, that the task difficulty has been rated just by the authors and not by independent raters. The three levels are only meant to distinguish the tasks and we do not have a specific hypothesis regarding this factor.

The experiment will be conducted within participants. Each participant will experience each modality condition in blocks, randomized in order. In each block all three demonstrations will be performed, again in randomized order. Participants will be instructed that purpose of the experiment is evaluating different feedback modalities in a teaching scenario, where the robot can see, hear and react to what they are doing but cannot make a conversation. They will be explained what to expect in the different conditions and that the robot knows about objects and actions.

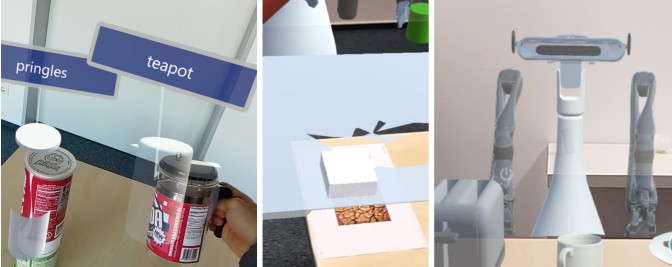

Fig. 3: Left: real object recognition via AR markers; Middle: AR marker anchoring the virtual robot in the real world; Right: the position and orientation of the virtual robot w.r.t. the teacher is used to control the reaction to teacher's behaviour.

### B. Setup

The system consists of two components: 1) front-end interface and 2) back-end state machine. Between the two components, a ROS-based communication channel is established to transfer human-input/system-output data (see Figure 2).

**Front-end interface**: virtual objects, a virtual robot, and XAI cues are displayed in the mixed-reality environment via the HoloLens[1]. HoloLens can scan the surroundings, build up 3D meshes of the environment objects and locate itself in the room, which enables it to stably overlay graphics in the environment considering occlusions with real objects. In this study, several fully virtual objects, such as a microwave, a cup, or a piece of bread were created and implemented in the AR world. Each object has a corresponding mass and collider, enabling human teachers to pick up, rotate and lay down them on real-world surfaces (e.g. table and floor, see Figure 1). Besides fully virtual objects, real objects can also be

---

[1]https://www.microsoft.com/en-us/hololens

recognized by the HoloLens via vuforia AR marker[2], allowing users to also manipulate those while teaching (Figure 3).

A virtual robot, which is the duplication of the physical robot "Johnny" [26], [27] with certain modifications, is integrated into the holographic environment (Figure 3 right). Same as the real world Johnny, the virtual robot's head can rotate along two axes (pitch and yaw). Two kinova arms[3] are also present in the virtual robot with the same degrees of freedom as the real ones. A vuforia AR-marker, which can be recognized by the HoloLens camera, is used to anchor the virtual robot in the physical space (Figure 3 middle).

Besides, hololLens is also responsible for detecting the human teacher's behaviour as input to the back-end system. This includes the teacher's head position/orientation, gaze-ray and speech input. More importantly, as holoLens can track the teacher's hand and fingers then the corresponding action (e.g., "pick" or "drop") can also be detected based on rule-based algorithms. Finally, the user behaviour and related manipulated object information is sent as ROS topics to the backend.

**Back-end state machine**: a finite state machine regulates the reactive behavior of the robot. Across all conditions, the system relies on the detection of certain multimodal events from the user to evaluate if the interaction is currently in a social context (e.g., during social attunement) or in a teaching context (during task demonstration). The robot (via the Hololens) can detect on which hologram the user is fixating, including the robot itself, recognize certain actions (picking, placing, switching on/off), recognize the utterance of known objects in the scene, as well as the user greeting ("Hello, Johnny!") or stating the end of the task ("Done"). The finite state machine regulating the robot's behavior consists hence of three states: idle, social coordination, and teaching. Switching from one state to another is regulated by two activation functions $\Phi_c$, related to each context $c = \{social, teaching\}$. Similarly to [28], the value of these functions is determined by a weighted sum of the currently detected social cues in the set $\mathcal{C}(m)$, for each modality in $\mathcal{M} = \{gaze, speech, action\}$:

$$\Phi_c(t) = \sum_{m \in \mathcal{M}} \sum_{i \in \mathcal{C}(m)} w_{mic} \cdot cue_i(t). \qquad (1)$$

where $cue_i \in \{0,1\}$, $\sum_m \sum_i w_{mic} = 1$. The weights are determined heuristically depending on the context. For example, in the social context, a fixation on the robot is weighted more than any utterance or action movement. In each state, the robot can display a gaze behavior, an XAI behavior, or both behaviors, depending on the experimental condition. Idle is the default state the robot is in when not interacting. Here, the robot randomly selects an object to focus on. As soon as the robot detects that the human is looking at it and addressing it $\Phi_{social}$ would raise over a certain threshold and also over $\Phi_{teaching}$, determining the switch to the social context. Here, the robot gazes back at the user (mutual gaze) in the gaze conditions or greets the user with an XAI cue

[2]https://developer.vuforia.com/
[3]armshttps://www.kinovarobotics.com/product/gen2-robots

in the XAI conditions. As the user starts the demonstration by looking at relevant objects and naming them, the robot switches to the teaching context, and focuses on the same objects as the tutor, either with its gaze or with the XAI cues. In the gaze condition, the first time an object is mentioned the robot repeats its name back, while in the XAI the object label appears directly on the object anytime the user looks at it.

### C. Data collection and analysis

With the Hololens and its connection to ROS, we plan to acquire timestamped data from the gaze and speech behavior of the interacting partners, the detected actions performed by the user, and the XAI cues displayed by the robot. As an objective measure, we consider overall demonstration times across feedback modalities, expecting shorter times in the XAI condition w.r.t. the gaze and speech condition (H1). In the combined condition, times might be longer but possibly still shorter than in the gaze condition (H2). As to behavioral measures, we consider the percentage of time spent by the user looking at the robot, expecting this to be shorter in the XAI/combined conditions. Conversely, the proportion of time spent looking at demonstration objects should be larger for the conditions with XAI cues. Further, we consider coordinated attention, that is, the time spent by user and robot on the same object or in mutual gaze (in the XAI only condition this would amount to the time the label is shown on the object). Here, the XAI conditions are expected to produce larger coordination, reducing the time spent by the user checking on the robot to follow the demonstration. Finally, after each condition block, participants will be administered the RoSAS questionnaire [29] to evaluate the social perception of the robot according to the three scales of *competence, warmth,* and *discomfort*. We expect the XAI condition to be rated higher on competence items, still, it might be possible that for the combined condition both competence and warmth are rated higher since gaze is a more human-like cue.

## V. DISCUSSION AND OUTLOOK

We presented a design for novel XAI cues in AR, to facilitate interactive learning by demonstration. Such interface is devised to solve the perceptual belief problem [9] making the tutor aware of the robot's current focus of attention, the label under which the robot knows the considered object and its state. Such cues also inform about the robot's understanding of the situation, i.e, recognizing actions and presenting a final report of the observed sequence. We reckon that this feedback modality should streamline demonstrations, allowing the user to build a mental model of the robot without explicit inquiry or longer observation. A pilot study will test the integration of all modules and tune the robot behavior heuristically in the different conditions, before proceeding with a user study. A positive evaluation of the XAI modality, also in integration with other modalities, will allow the development of a pragmatic frame [8], i.e., a teaching interaction protocol, where both tutor and learner share a common ground and the user can tailor the demonstration on the learner knowledge and capabilities.

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
