# OpenReview forum: "Investigating explainable human-robot interaction with augmented reality"
_humanrobotinteraction.org/HRI/2022/Workshop/VAM-HRI — VAM-HRI 2022_

### Official Review · Reviewer_k5d1 · 2022-02-24
**Welcome contribution to the workhsop.**

**Rating:** 8
**Confidence:** 5

**Review:**

The authors provide a method for explainable learning with demonstration using augmented reality.  The work is a welcome contribution to the workshop.

---

### Official Review · Reviewer_fyr2 · 2022-02-28
**Application of XAI for VAMHRI, accept**

**Rating:** 8
**Confidence:** 5

**Review:**


This paper describes a technique for providing XAI information during task training activities, along with a human subjects study to evaluate it. The technique includes object recognition, action recognition, and simple voice recognition, along with associated visual labels displayed in AR. The robot and objects are also virtually displayed. A state machine is also described that regulates the robot's behaviors and actions. A pilot study will be performed first, then a full scale human subjects study. This is an interesting line of research in a field ripe for VAMHRI. Please see a few questions and suggestions below.

- You may wish to see how [1] and [2] could inform your approach. These relevant works may further guide your study.
- How are the limited states and actions chosen for the state machine? In looking at Figure 1, it appears that these are pre-set options; how were/are they decided?
- Is the task ever demonstrated back to the participant? How might this affect the user's perception of the system?

Minor edits:
- Ensure that the word "HoloLens" is correctly typed and capitalized in all uses.
- Spell out "with respect to" in IV C.

[1] E. Rosen, D. Whitney, E. Phillips, G. Chien, J. Tompkin, G. Konidaris and S. Tellex, "Communicating and controlling robot arm motion intent through mixed-reality head-mounted displays," The International Journal of Robotics Research. 2019, 38(12-13): 1513-1526. doi: 10.1177/0278364919842925.
[2] M. B. Luebbers, C. Brooks, C. L. Mueller, D. Szafir and B. Hayes, "ARC-LfD: Using Augmented Reality for Interactive Long-Term Robot Skill Maintenance via Constrained Learning from Demonstration," 2021 IEEE International Conference on Robotics and Automation (ICRA), 2021, pp. 3794-3800, doi: 10.1109/ICRA48506.2021.9561844.

---

### Decision · Program_Chairs · 2022-03-04

Accept